# Use of Drugs for ATTRv Amyloidosis in the Real World: How Therapy Is Changing Survival in a Non-Endemic Area

**DOI:** 10.3390/brainsci11050545

**Published:** 2021-04-27

**Authors:** Massimo Russo, Luca Gentile, Vincenzo Di Stefano, Gianluca Di Bella, Fabio Minutoli, Antonio Toscano, Filippo Brighina, Giuseppe Vita, Anna Mazzeo

**Affiliations:** 1Unit of Neurology and Neuromuscular Diseases, Department of Clinical and Experimental Medicine, University of Messina, 98122 Messina, Italy; lucagentile84@yahoo.it (L.G.); antonio.toscano@unime.it (A.T.); giuseppe.vita@unime.it (G.V.); annamazzeo@yahoo.it (A.M.); 2Department of Biomedicine, Neuroscience and Advanced Diagnostic (BIND), University of Palermo, 90133 Palermo, Italy; vincenzo19689@gmail.com (V.D.S.); filippo.brighina@unipa.it (F.B.); 3Cardiology Unit, Department of Clinical and Experimental Medicine, AOU Policlinico G. Martino, University of Messina, 98122 Messina, Italy; gianluca.dibella@unime.it; 4Department of Biomedical and Dental Sciences and Morphofunctional Imaging, University of Messina, 98122 Messina, Italy; fabio.minutoli@unime.it

**Keywords:** hereditary transthyretin amyloidosis, ATTRv, non-V30M, polyneuropathy, survival, tafamidis, patisiran, inotersen

## Abstract

Background: Over the past decade, three new drugs have been approved for the treatment of hereditary amyloid transthyretin (ATTRv) polyneuropathy. The aim of this work was to analyze whether current therapies prolong survival for patients affected by ATTRv amyloidosis. Methods: The study was conducted retrospectively, analyzing the medical records of 105 patients with genetic diagnoses of familial amyloidotic polyneuropathy followed at the two referral centers for the disease in Sicily, Italy. Of these, 71 received disease-modifying therapy, while 34 received only symptomatic treatment or no therapy. Results: The most used treatment in our patient cohort was tafamidis, followed by liver transplantation, patisiran, inotersen, and diflunisal. The median survival was significantly longer for treated vs. untreated patients (12 years vs. 8 years). In the 71 patients who received disease-modifying treatment, the presence of cardiac involvement, weight loss, or autonomic dysfunction at diagnosis was not related to survival. Conversely, patients diagnosed in the early stage of the disease (PND 1) had significantly longer survival than those diagnosed in the late stage (PND 2–4).

## 1. Introduction

Hereditary amyloid transthyretin (ATTRv) amyloidosis with polyneuropathy is a systemic adult-onset disease that affects the sensorimotor and autonomic functions along with other organs (especially the heart, gastrointestinal tract, eyes, and kidneys) [1]. The disease has an autosomal dominant pattern of inheritance and is considered a worldwide disease with an estimated global prevalence of 38,000 persons [2,3,4,5]. Patients with the most frequent neuropathic mutation, V30M (p.V50M), have characteristically early onset (<50 years) in endemic areas, whereas late onset (>50 years) is prevalent in non-endemic areas. Other variants, such as T49A (p.T69A) and E89Q (p.E109Q), often present with an early onset even in non-endemic areas [6,7,8,9,10]. The most common hereditary cardiac transthyretin amyloidosis is that associated with V122I (p.V142I), carried by 3.9% of individuals of African descent [11].

TTR is a transport protein in the serum and cerebrospinal fluid that carries the thyroid hormone thyroxine and retinol-binding protein bound to retinol. It is encoded by the *TTR* gene located in the 18th chromosome. The protein is a 55kDa homotetramer synthesized in the liver, choroid plexus, and retinal pigment epithelium for secretion into the bloodstream, cerebrospinal fluid, and eye, respectively. Each monomer is a 127-residue polypeptide rich in beta sheet structure [12,13,14]. The presence of missense mutations in altering the amino acid sequence can make the tetramer less stable, favoring its dissociation. The misfolded monomers aggregate, generating amyloid fibrils, which precipitate into tissues [15,16]. In peripheral nerves, fibrils cause axon and Schwann cell damage, resulting in the predominant loss of small-fiber axons characteristic of early-onset cases, while vasculopathy can also determine the pathogenesis of neuropathy in late-onset cases [17]. Once polyneuropathy begins, it progresses rapidly, evolving in three defined stages: in the first, patients have a sensory polyneuropathy that leads to difficulty walking without assistance; in the second, assistance with walking is required; and in the last stage, patients are wheelchair-bound or bedridden. Death occurs on average 12 years after the onset of the disease in the absence of therapy, but survival can vary with different mutations [18]. The presence of amyloidotic heart disease can influence this classical progression, and the survival for variants with prevalent cardiological involvement is generally shorter [19,20]. The clinical phenotype based on the mutations is not always predictable, as there is considerable variability in clinical expression even within the same family [21].

In the last three decades, there has been an overall improvement in the management of this disease, and various disease-modifying therapies have been developed [15,16]. Moreover, awareness of being ahead of a multisystem pathology requiring a multidisciplinary approach has also improved the management of symptoms [18,22]. In parallel, increased diagnostic tools and awareness of the disease have reduced the diagnostic delay, favoring an earlier start of treatment [4,23].

To date, some studies on the efficacy of new drugs in modifying the outcomes of patients over short periods have been performed, but information on survival is lacking, mainly in non-V30M patients [24,25,26,27,28,29,30,31,32]. A recent paper showed that in non-endemic late-onset ATTRv amyloidosis, long-term treatment with tafamidis is safe and efficacious, and the severity of polyneuropathy at the start of treatment is the main predictor for disease progression on tafamidis [30].

The aim of this work was to analyze whether current therapies have prolonged survival in Sicilian patients affected by ATTRv amyloidosis, representing an almost exclusively non-V30M patient population. Another purpose was to update the clinical–epidemiological data in Sicily [10], where diagnosed cases have increased dramatically in the last five years.

## 2. Materials and Methods

We conducted a retrospective study, including all patients and carriers who had a pathogenic missense mutation on the *TTR* gene followed in the two Sicilian referral centers for the disease (Amyloidosis Centre of Messina University Hospital and Amyloidosis Centre of Palermo University Hospital) from January 1991 until December 2020. The study was approved by the Ethical Committee of Messina Province. All methods were performed in accordance with the relevant guidelines and regulations. The diagnosis was based on clinical and neurophysiological features and confirmed by positive genetic testing for *TTR* gene variants. In some patients, usually the probands, especially in the first half of the observed period, the diagnosis was also corroborated by biopsy, in most cases of the sural nerve. We also performed *TTR* gene analysis in asymptomatic relatives close to the typical age of onset, with the aim of following them serially to reduce the diagnostic delay and, consequently, to start therapy in a timely manner.

The clinical data collected from available medical records consisted of:The age at onset, diagnosis, and death, or age of living patients as of 31 December 2020,The family history of neuropathy or ATTR amyloidosis,The type of *TTR* mutation,The clinical and neurological examination at diagnosis,Electromyography and nerve conduction studies at diagnosis,The presence of autonomic nervous system involvement at diagnosis, when available and based on autonomic cardiovascular tests, or, in their absence, based on clinical evaluation,The presence of cardiac involvement at diagnosis based on ECG, echocardiogram, atrial natriuretic hormone dosage (nt-pro-BNP), and, in some cases, 99 m Tc-3, 3-diphosphono-1, 2-propanodicarboxylic acid scintigraphy or cardiac magnetic resonance (CMR) with T2-weighted imaging and late gadolinium enhancement,Sural nerve biopsy in selected cases,Therapeutic drugs (or procedures) and the overall duration of their administration,Symptomatic treatment.

### Statistical Analysis

The characteristics of the patients were described by using standard descriptive statistics. All the results are documented as mean values, ranges, or percentages when appropriate. To assess the relationships between treatment and survival, between clinical characteristics in treated patients at the onset and survival, and between mutation (E89Q vs. F64L) and survival, curves were estimated by Kaplan–Meier analysis and compared using the log rank test. The effect size is expressed as the hazard ratio, and statistical uncertainty is expressed as 95% confidence intervals. The comparison of the age of onset and disease duration for deceased patients between treated and untreated was performed using chi squared tests.

## 3. Results

### 3.1. General Aspects

This study examined 159 patients with pathogenic mutations in the *TTR* gene at the two reference centers from 1991 to 2020, 135 in Messina and 24 in Palermo. The most frequently encountered mutation was F64L (p.F84L) in 81 patients, followed by E89Q in 56, V122I (p.V142I) in 11, T49A in 8, and V30M in 3. Among them, 40 subjects (20 men and 20 women, mean age 46.6, range 28–66) were still asymptomatic as of 31 December 2020. In addition, 14 patients (4 men and 10 women, mean age 58.3 years, range 41–78) were affected by carpal tunnel syndrome, without other symptoms or signs of ATTRv amyloidosis. The clinical characteristics of the other 105 patients with ATTRv amyloidosis are described in Table 1. Of the 105 patients, 71 received disease-modifying therapy and 34 did not receive specific drugs for the disease.

Three mutations usually have late onset (F64L, V122I, and V30M), T49A has early onset, and the age of onset of patients with the E89Q variant is usually around 50. Clinical evaluations at diagnosis show frequent cardiac involvement in E89Q and V122I patients, while dysautonomia is frequently present in T49A patients. New diagnoses progressively increased in the 30-year period, reaching 45 cases in the last five years (Figure 1). The most frequently identified variant over the past five years was F64L, in 29 of 45 patients.

Regarding the type of polyneuropathy, in the 105 patients who developed polyneuropathy, a nerve conduction study at diagnosis showed an axonal pattern in 94, demyelinating in 5, and mixed in 6. Sural nerve biopsy was performed in 16 patients, among which 10 were positive to Congo Red staining; in all of these cases there was a marked loss of myelinated fibers.

### 3.2. Actual Scenario

As of 31 December 2020, 73 living patients and 40 asymptomatic carriers underwent regular follow-up, and 44 patients received disease-modifying treatment: 23 patisiran (3 also received liver transplants), 17 tafamidis, and 4 inotersen. Only one patient was liver-transplanted and did not receive other therapies. No other disease-modifying treatments are currently in use in Sicily. In light of these results, the current prevalence of ATTRv amyloidosis in Sicily is 15.0/1,000,000 (12.1/1,000,000 excluding patients with carpal tunnel syndrome only).

### 3.3. Comparison of Treated and Untreated Patients

Table 2 shows the characteristics of patients who received a disease-modifying treatment, compared to those patients who did not.

The two groups did not show a significant difference in age of onset (*p* = 0.24) and were quite similar, considering patient *TTR* mutations and symptoms at diagnosis. The type of treatment administered was measured as years of therapy received, and in the case of patients who underwent liver transplantation (six in this cohort), the years of therapy were considered as all years of life after the transplant. The Kaplan–Meyer curves of survival in the two groups are represented in Figure 2. The survival was longer for treated patients (*p* = 0.0004).

The median survival of treated and untreated patients was 12 years vs. 8 years, and the hazard ratio for untreated/treated was 2.55. Furthermore, considering only deceased patients, the disease duration was longer for treated patients than untreated patients (*p* = 0.04) (Table 2).

### 3.4. Relation between Clinical Characteristics at the Onset and Survival in Treated Patients

In the 71 treated patients, survival was not significantly different as it related to the presence of cardiac involvement (*p* = 0.79), weight loss (*p* = 0.1), or autonomic dysfunction (*p* = 0.05) at diagnosis. Furthermore, survival was not significantly different comparing patients with E89Q vs. those with F64L mutation (*p* = 0.19). A very significant difference (*p* = 0.0002) was found between patients who had a polyneuropathy disability score (PND) [34] of 1 (sensory disturbance but preserved walking capability) at diagnosis vs. patients with a more advanced polyneuropathy score (Figure 3).

### 3.5. Symptomatic Drugs

In 51 of the 105 patients with polyneuropathy, a specific treatment for neuropathic pain had been administered. However, these findings are likely to have been underreported in the clinical notes, especially in the first half of the period under review, because the general practitioner sometimes managed these drugs. The most used drugs for neuropathic pain were pregabalin (38 cases), gabapentin (4), amitriptyline (21), duloxetine (12), and tramadol (6). The most used symptomatic treatments for autonomic symptoms were fludrocortisone, midodrine, loperamide, and octreotide. Cardiac dysfunction was generally treated with diuretics, ace inhibitors, beta blockers, and implantable cardioverter defibrillators. No patients underwent a heart transplant.

## 4. Discussion

### 4.1. Epidemiological Aspect

We reported long-term data on a cohort of patients with genetically confirmed ATTRv amyloidosis followed at the two referral centers for amyloidosis in Sicily, the University Hospitals of Messina and Palermo, between January 1991 and July 2020.

In the last five years, the increased attention to this disease has led to a significant increase in diagnoses, now almost double those in the previous five years [10]. These findings, together with the lengthening of disease duration, also show a significant increase in prevalence compared to the 2015 data (15 vs. 8.8 per million), the highest in Italy and one of the highest in Europe among non-endemic areas [4,5].

Patients diagnosed in the last five years were mainly carriers of the F64L mutation. This variant has been evidently underdiagnosed, given the advanced age at onset, the almost exclusively neuropathic phenotype, and the high frequency of apparently sporadic cases [9]. Furthermore, two new mutations were detected in the Sicilian cohort: the well-known V30M with the typical neuropathic phenotype, described in non-endemic areas, and the V122I mutation (in six patients), commonly described as causing an almost exclusive cardiac phenotype [35] that, in our experience, instead determined in all cases a neuropathic phenotype, with cardiac amyloidosis in five of them [21,36].

### 4.2. Diagnostic Delay and Carpal Tunnel Syndrome

The diagnostic delay has been significantly reduced in recent years [4]. For this reason, in the group of untreated patients mainly in the period before 2011 (23 of 34), the diagnostic delay was significantly higher (3.5 vs. 2.5 years). Conversely, most of the treated patients were diagnosed after 2011 (54 of 71), the year the European Medical Agency (EMA) approved the first drug for ATTRv amyloidosis, tafamidis meglumine, which led to increased medical awareness [24,25,29].

Among the 113 subjects carrying pathogenic mutations who underwent regular follow-up, 40 were asymptomatic carriers, 59 had polyneuropathy and/or involvement of other organs, and 14 had only carpal tunnel syndrome. Although there was no indication to start treatment in the latter group, recent studies have shown that in most biopsies obtained from the transverse carpal ligament of such patients, amyloid is present [37,38]. In our opinion, the possibility that therapy with TTR stabilizers or *TTR* mRNA silencers could delay the onset of systemic amyloidosis in such patients should be evaluated. While waiting for further clarification on this topic, we will perform frequent clinical exams (three times a year), neurophysiological studies (NCS and cardiovascular autonomic testing, one to two times a year), and imaging studies (cardiac MRI and/or serial scanning with 99 m Tc-3, 3-diphosphono-1, 2-propanodicarboxylic acid once a year) to promptly start treatment [39,40,41].

### 4.3. Treatment Used

The percentage of patients receiving treatment has increased considerably, owing to the availability of tafamidis, approved by the EMA in November 2011 for adult patients with transthyretin-related polyneuropathy. This drug acts as a pharmacologic chaperone that stabilizes the properly folded tetramer of TTR protein by binding in one of the two thyroxine-binding sites [42]. The U.S. Food and Drug Administration (FDA) in 2019 and the EMA in 2020 approved tafamidis also for the treatment of transthyretin-mediated cardiomyopathy (ATT-CM) [43]. Although it is not always able to alter the disease, it is certainly very well tolerated and manageable [29].

Almost exclusively non-Met30 patients composed our cohort, therefore in some cases we preferred to shift patients to the new oligonucleotide drugs that have been shown to be unmistakably effective even in this population. Overall, both drugs, and particularly patisiran, have demonstrated better efficacy compared to tafamidis [27,28,44,45,46]. Despite this, in 17 of 44 patients, tafamidis was still used based on the abovementioned advantages.

The long-term survival after liver transplantation is encouraging in patients with early-onset V30M, but the efficacy is controversial in non-Met30 patients, especially when associated with lower body mass index or older age, which was probably the reason why only six patients underwent liver transplantation in our cohort [34,47,48,49]. Different studies have explained why many patients continue to worsen after liver transplant. Tissue accumulation of TTR can continue, likely because TTR amyloid fibers promote subsequent deposition of wild-type TTR [50]. The ability of amyloid seeding to promote fibrillogenesis could be particularly relevant in patients with cardiac amyloid deposits. In recent years, to inhibit this process, peptides designed to complement the structures of TTR fibrils inhibiting TTR fibril seeding have been tested. However, these molecules have not been verified in clinical phase studies [15,51,52]. Patients with a cardiac or mixed (cardio/neuro) phenotype, as often happens in ATTRE89Q amyloidosis, who have cardiac amyloid deposits at the time of diagnosis, would require combined heart–liver transplant to avoid the seeding effect. However, the exclusion of older patients and those with severe systemic disease, limited organ availability, and the risks associated with surgery and life-long immunosuppression represent difficulties that often discourage clinicians from proposing combined transplantation [53]. In our experience, in some cases patients rejected the transplant option because a close relative who underwent transplantation had a negative outcome. Finally, in patients with ATTRF64L amyloidosis, the indications for transplantation are very limited due to the advanced age at diagnosis. It is therefore not surprising that the most commonly used therapy in Sicily in the period under observation was tafamidis (despite its only being available since 2011) rather than liver transplantation. Patisiran and inotersen will probably be the most commonly used drugs in the coming years, awaiting further pharmacologic developments.

### 4.4. Effect on Survival

The survival analysis performed in this study was an expression of the effects of different treatments, which in some cases were used at different times in the same patient and in others even simultaneously (for example, in the three transplant patients currently taking patisiran). Obviously, this limitation of the study made it possible to formulate an overall judgment only on the progress achieved in ATTRv amyloidosis.

Overall, the major role in prolongation in terms of the survival of treated patients was mainly played by tafamidis, which was the treatment used most widely and for the longest period in this cohort by the number of years (192 years of treatment, 3.7 years per patient). In descending order, the other therapies were liver transplant (52 years, 8.7 years per patient), patisiran (34 years, 1.9 years per patient), inotersen (14 years, 2 years per patient), and diflunisal (only 4 years altogether, 2 years per patient) (Table 2).

The survival curves between treated and untreated patients show a clear difference 8–9 years after the onset of symptoms. Disease duration in deceased patients was also significantly different. This difference could become even greater in the coming years thanks to the new treatments.

In this study, there was no prognostic difference in the patients treated in relation to the presence of cardiac involvement, autonomic dysfunction, or malnutrition at diagnosis [30]. This unexpected finding could be, in part, because mutations with a higher prevalence of cardiac and autonomic involvement at diagnosis may cause disease onset at a younger age, and consequently in patients with much longer life expectancy. As expected, when survival is related to the PND score at diagnosis, patients in stage 1 have a higher rate of survival. The patients in the study cohort were diagnosed in almost all cases in a neurological clinic; therefore, the presence of neurological symptoms at onset was constant in almost all cases. Consequently, the presence of polyneuropathy at diagnosis does not discriminate between variants or depend on the age at onset.

## 5. Conclusions

As predicted in previous reports [9,10], the prevalence of transthyretin amyloidosis is increasing significantly in Sicily owing to greater attention being paid to this pathology. Disease-modifying therapies, in percentage mainly represented by tafamidis at present, have significantly improved the survival of patients. When patients are diagnosed with a PND score of 1, their overall survival is longer.

## Figures and Tables

**Figure 1 brainsci-11-00545-f001:**
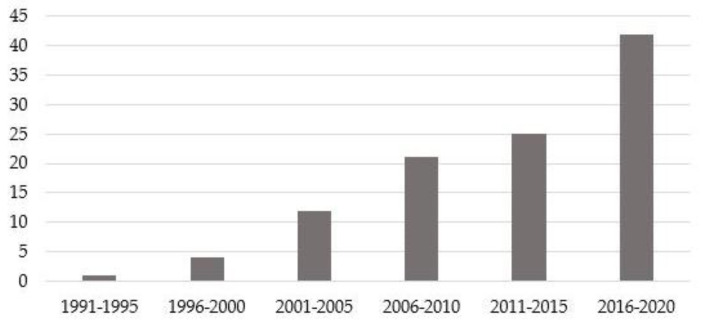
Number of diagnoses for five-year periods.

**Figure 2 brainsci-11-00545-f002:**
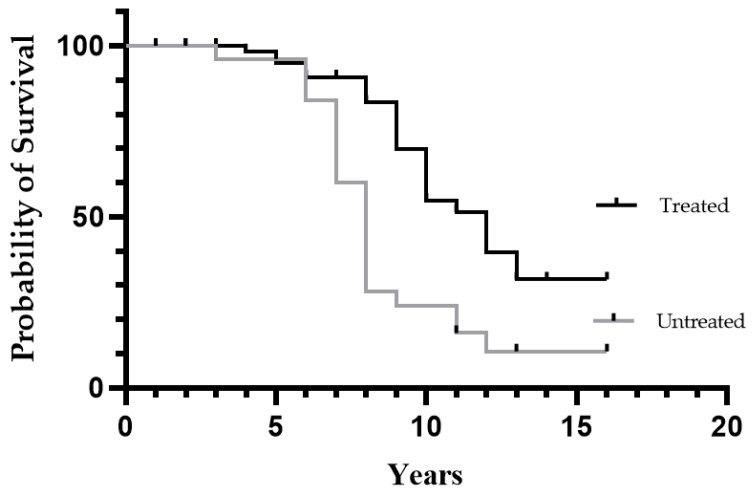
Survival comparing patients who received disease-modifying treatment (*n* = 71) vs. untreated patients (*n* = 34).

**Figure 3 brainsci-11-00545-f003:**
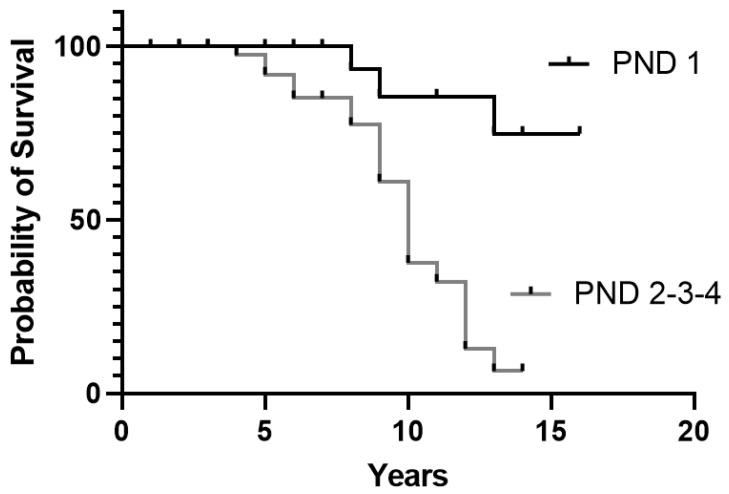
Survival according to PND score at diagnosis for 71 treated patients: 1 = sensory disturbance; 2 = impaired walking capability but ability to walk without a stick or crutches; 3 = walking with the help of one stick or crutch; 4 = confined to wheelchair or bedridden.

**Table 1 brainsci-11-00545-t001:** Clinical characteristics. All time intervals are expressed in years.

	Total	F64L	E89Q	V122I	T49A	V30M
Patients, *n* (m/f)	105 (68/37)	51 (36/15)	38 (21/17)	6 (6/0)	8 (3/5)	2 (2/0)
Age at onset, mean (range)	59.3 (33.0–81.2)	66.6 (46.3–81.2)	51.7 (36.4–76.2)	65.3 (62.3–69.1)	43.6 (33.0–56.3)	62.4 (53.7–71.1)
Age at diagnosis, mean (range)	62.2 (33.9–85.3)	70.1 (52.3–85.3)	53.5 (40.2–74.8)	70.1 (66.0–76.1)	45.4 (33.9–57.7)	65.9 (56.4–75.3)
Diagnostic delay (years), mean (range)	2.8 (0–16)	3.5 (0–16)	1.9 (0–6)	4.8 (3–7)	1.75 (0–4)	3.5 (3–4)
Probands, *n* (%)	50 (47.6)	35 (68.6)	7 (18.4)	1 (16.7)	1 (12.5)	0
Late onset, *n* (%)	81 (77.1)	50 (98.0)	22 (57.8)	6 (100)	1 (12.5)	2 (100)
Neurologic symptoms at diagnosis, *n* (%)	103 (98.0)	51 (100)	36 (94.7)	6 (100)	8 (100)	2 (100)
Sensory	97 (92.4)	45 (88.2)	36 (94.7)	6 (100)	8 (100)	2 (100)
Motor	60 (57.1)	37 (72.5)	15 (39.5)	0	6 (75)	2 (100)
Gait difficulties	46 (43.8)	32 (62.7)	11 (28.9)	0	1 (12.5)	2 (100)
CTS	62 (59.0)	30 (58.8)	24 (63.2)	4 (66.7)	3 (37.5)	1 (50)
Cardiac involvement at diagnosis, *n* (%)	48 (45.7)	9 (17.6)	31 (81.6)	5 (83.3)	3 (37.5)	0
Dysautonomia at diagnosis, *n* (%)	60 (57.1)	24 (47.1)	26 (68.4)	1 (16.7)	8 (100)	1 (50)
CADT score point loss	5.5	6.2	5.1	6	5.2	2
Weight loss, *n* (%)	53 (50.5)	25 (49.0)	16 (42.1)	1 (50)	5 (87.5)	2 (100)
Living patients, *n* (%)	59 (56.1)	27 (52.9)	23 (60.5)	3 (50)	4 (50)	2 (100)
Disease duration in living pt. (yrs)	6.15 (0.4–16)	4.8 (0.4–16)	6.8 (0.5–13.6)	8.9 (4.9–13.6)	7.3 (5.5–8.6)	9.6 (5.5–13.7)
Deceased patients, *n* (%)	46 (43.8)	24 (47.0)	15 (39.4)	2 (33.3)	5 (62.5)	0
Disease duration in deceased pt. (yrs)	8.4 (3.1–13.0)	8.5 (4.0–11.8)	8.2 (3.1–13.0)	7.5 (6.3–8.7)	9.2 (7.0–12.9)	N.A.

Late onset: >50 years. Sensory: abnormal pinprick or vibration or joint position sense at neurological examination. Motor: Medical Research Council scale ≤4 due to polyneuropathy in two or more muscles. Gait recorded as abnormal due to polyneuropathy at neurological examination. Cardiac involvement at diagnosis: presence of cardiac amyloidosis demonstrated by echocardiography, scintigraphy, or cardiac resonance. Dysautonomia at diagnosis: presence of postural hypotension, gastrointestinal, or genito-urinary dysfunction due to autonomic failure based on clinical judgment. Weight loss: number of patients with unexplained body weight reduction ≥5 kg since onset of symptoms. CTS, carpal tunnel syndrome; CADT, Compound Autonomic Dysfunction Test [33]; N.A., not applicable.

**Table 2 brainsci-11-00545-t002:** Comparison of treated vs. untreated patients and details of treatments received.

	Treated		Untreated	*p*
Patients, *n* (m/f)	71 (51/20)		34 (16/18)	
Age at onset, mean (range)	58.5 (33.0–81.2)		61.0 (37.5–75.7)	0.24
Diagnosis before 2011, *n* (%)	17 (43)		23 (57)	
Diagnosis after 2011, *n* (%)	54 (83)		11 (17)	
Type of mutation: F64L	30		21	
E89Q	28		10	
T49A	6		2	
V122I	5		1	
V30M	2		0	
Neurologic symptoms at diagnosis, *n* (%)	69 (97)		34 (100)	
Cardiac involvement at diagnosis, *n* (%)	37 (52)		11 (32)	
Dysautonomia at diagnosis, *n* (%)	43 (60)		17 (50)	
Weight loss at diagnosis, *n* (%)	39 (55)		14 (41)	
Years of disease, total (mean)	531 (7.48)		220 (6.47)	
Years of treatment received (*n* patients/mean)	299	(71/4.2)		
Tafamidis	192	(52/3.7)		
Liver transplantation	52	(6/8.7)		
Patisiran	37	(20/1.9)		
Inotersen	14	(7/2)		
Diflunisal	4	(2/2)		
Diagnostic delay, total (mean)	180 (2.5)		119 (3.5)	
Time out of treatment for other reasons	52		101	
Deceased patients, *n* (%)	24 (34)		22 (65)	
Disease duration in deceased patients, mean (range)	9.1 (4–13)		7.7 (3–12)	**0.04**

Significant *p*-values are in bold.

## Data Availability

The data presented in this study are available on request from the corresponding author.

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
