# Peer review of "Use of Drugs for ATTRv Amyloidosis in the Real World: How Therapy Is Changing Survival in a Non-Endemic Area"

_brainsci, 2021, doi:10.3390/brainsci11050545_

Round 1
Reviewer 1 Report
The authors present an important study comparing the effect of disease-modifying treatment on mainly the survival of ATTRv amyloidosis patients in a non-endemic area. The study is of particular interest since most other studies are carried out on ATTRV30M amyloidosis. It is also important to continuously evaluate the effects of the recently approved therapies for ATTRv amyloidosis, especially for patients with non-V30M mutations. I only have few minor comments to the authors:
- For clariy, please also refer to the full precursor for each TTR mutation (e.g. p.V50M) at least once in the manuscript.
- Please refer to the TTR gene in italic.
- Material and methods, row 92: Please refer to the disease as ATTR amyloidosis.
- Material and methods, rows 99-100: Please clarify if cardiac involvement was evaluated with nt-pro-BNP or another lab test and specify the type of scintigraphy and resonance that was used.
- Material and methods, row 104: Please consider updating this term to symptomatic treatment.
- Results, Table 1: Please explain the CADT score.
- Results, 3.3: If possible, please consider adding a comparison of patients diagnosed in different time periods (e.g. 1991-2001, 2001-2011 and 2011 to 2020) to see if also the general treatment and follow-up of the patients may have contriuted to the improved survival.
Reviewer 2 Report
This is an interesting review of non-V30M patients treated at 2 Sicilian referral centers. The findings and conclusions are consistent with overall literature.
My suggestions are to significantly shorten the paper, especially by removing the speculative and predictive aspects, to clarify the abbreviations and to improve some of the grammar.
Reviewer 3 Report
Page 1 lines 6-7 re-format
Line 37: TTR V30 M is the most frequent neuropathic mutation worldwide TTRV122I is probably most common clinically relevant variant
Line 52: should read "Once polyneuropathy begins it progresses rapidly, evolving in three defined stages"
Line 55: death in 12 years average is in the absence of therapy and may vary with different mutations, even with same mutation.
Lines 68-71: there other references that address the issue of survival:
e.g. Coelho T et al J. Neurology 2018
Merlini et al Neurologic Therapy 2020
Ungerer MN et al Amyloid 2020
In table 1 the units for some parameters are not stated e.g. for diagnostic delay are the units months or years, is weight loss in kg or lbs.
Table 2: I do not think that total years of treatment or total years of disease is helpful. The mean is OK, the same is true for diagnostic delay. The latter is critical since all the therapeutic data to date indicate that the best results are obtained when treatment is started early in the course of disease. Clinical neuropathy staging does not always correlate precisely with duration of disease, i.e. not all stage I's are of the same duration, with some responding very well and others not (in the tafamidis study), perhaps related to delay in diagnosis and the relative coarseness of peripheral neuropathy staging. While disease duration in deceased treated and untreated patients differs with a p value of 0.04 I am not sure that is meaningful in non-simultaneous cohorts. It would be nice to see Kaplan-Meier for F64L and E89Q patients done separately, although the sample sizes might not be adequate to see statistically significant differences, but it might compensate for differences lost in the pooling of all the mutations. If the data differ it might support the notion of the heterogeneity of different mutations in their response to treatment.
Is diagnostic delay in months or years?
Line 176: the failure to see a difference in survival with cardiac involvement could suggest that these patients may have been later in the course of their disease than was thought. Was there a correlation between cardiac involvement and degree of neuropathy? If not, it might suggest that the techniques for defining cardiac involvement are more sensitive than those for diagnosing polyneuropathy, although symptoms are not.
I agree with the notion of considering therapy in patients with CTS only in a preventive mode, but why stop there? Why not treat the mutation since it appears that the earlier treatment is begun the better the chances for success. We have at least one agent that is effective, and safe, although not yet cheap enough for a true prevention study in allele carriers.
I am not certain that at this point we can say that oligonucleotides are better than tafamidis. All the RCT's indicate that about 2/3 of patients respond, some better than others, 1/3 do not (see Monteiro et al as well as your reference 23). Hence your comments in lines 269-70 are probably premature regarding the ultimate usage of the oligonucleotides, particularly inotersen with its much higher incidence of side effects in its current formulation, as well as the cost. It is possible that the new formulations currently in trials will give us better outcomes and less frequent dosing.
The recently described CNS effects late after liver transplantation in V30M patients are disturbing since none of the current therapies cross the BBB. Can you comment on whether there were CNS findings in your patients during long term follow up as reported by Sekijima and Maia in their series.
